# Microbial Metabolic Limitations and Their Relationships with Sediment Organic Carbon Across Lake Salinity Gradient in Tibetan Plateau

**DOI:** 10.3390/microorganisms13030629

**Published:** 2025-03-11

**Authors:** Weizhen Zhang, Jianjun Wang, Yun Li, Chao Song, Yongqiang Zhou, Xianqiang Meng, Ruirui Chen

**Affiliations:** 1Center for Pan-Third Pole Environment, Lanzhou University, Lanzhou 730000, China; 2Chayu Monsoon Corridor Observation and Research Station for Multi-Sphere Changes, Xizang Autonomous Region, Chayu 860600, China; 3State Key Laboratory of Lake Science and Environment, Nanjing Institute of Geography and Limnology, Chinese Academy of Sciences, Nanjing 210008, China; jjwang@niglas.ac.cn (J.W.); liyun@niglas.ac.cn (Y.L.); yqzhou@niglas.ac.cn (Y.Z.); xqmeng@niglas.ac.cn (X.M.); 4University of Chinese Academy of Sciences, Beijing 100049, China; 5State Key Laboratory of Herbage Improvement and Grassland Agro-Ecosystems, College of Ecology, Lanzhou University, Lanzhou 730000, China; chaosong@lzu.edu.cn; 6College of Chemical Engineering, Nanjing Forestry University, Nanjing 210008, China; rrchen@njfu.edu.cn

**Keywords:** extracellular enzyme, microbial metabolic limitation, salinity, pH, sediment organic carbon

## Abstract

Inland lakes, contributing substantially to the global storage of sediment organic carbon (SOC), are subject to marked changes in salinity due to climate warming. The imbalance in the supply of resources, such as carbon, nitrogen, and phosphorus, in sediments leads to microbial metabolic limitations (MMLs). This, in turn, triggers the secretion of extracellular enzymes by microorganisms to mine for deficient resources by decomposing complex organic carbon. This process is a rate-limiting step in the degradation of organic carbon and, as a result, has the potential to regulate organic carbon stocks. However, the general understanding of MML patterns and their relationships with SOC content along lake salinity gradients remains elusive. This study examined 25 lakes on the Tibetan Plateau with salinity ranging from 0.13‰ to 31.06‰, analyzing MMLs through enzymatic stoichiometry. The results showed that sediment microbial metabolism was mainly limited by carbon and nitrogen, with stronger limitations at higher salinity. Water salinity and sediment pH were the main factors influencing microbial limitations, either directly or indirectly, through their effects on nutrients and microbial diversity. Additionally, the SOC content was negatively correlated with microbial carbon limitation, a relationship weakened when salinity and pH were controlled. These findings suggest that the decrease in SOC with increased salinity or pH could be driven by stronger microbial carbon limitations, offering insights into the impact of salinity changes on SOC stocks in inland lakes due to climate change.

## 1. Introduction

Increasing climate warming is causing significant changes in the salinity of inland lakes [1], leading to either salinization [2] or desalinization [3]. Specifically, the water salinity of high-altitude lakes on the Tibetan Plateau declined as lake areas expanded and water storage increased due to the warming and wetting climate from the 1970s to the 2010s [3]. In contrast, many saline lakes worldwide, located in arid and semi-arid regions such as Lake Urmia in Central Asia and the Great Salt Lake in the United States, have been suffering from desiccation and shrinking at drastic rates, resulting in considerable increases in lake salinity over the past several decades [2,4]. Lake sediments, which store large amounts of organic carbon [5], are critical in understanding how salinity changes affect these reserves [6,7,8]. Microorganisms in sediments play pivotal roles in the turnover of sediment organic carbon (SOC) [9], but our understanding of their relationship with SOC dynamics under varying salinity conditions is limited, hindering accurate SOC estimation.

Salinity can alter microbial metabolism, impacting organic carbon turnover in lake sediments. High salinity induces dehydration, protein inactivation, and ion imbalance in microbes [10], forcing them to accumulate compatible solutes, including carbohydrates and amino acid derivatives, for osmotic balance or energy [11]. When salinity decreases, microbes metabolize these solutes into nutrients, dissolved organic carbon, and carbon dioxide to avoid cytolysis [12]. Additionally, salinity changes can reduce microbial diversity and shift dominant species, affecting carbon metabolism [13,14]. For instance, strains within one genus (e.g., *Pseudoalteromonas* and *Halomonas*) isolated from low-salinity lakes demonstrate higher carbon utilization abilities than their counterparts from high-salinity lakes [15]. Increased salinity also reduces nutrient availability in sediments, affecting microbial access to essential nutrients like phosphate and ammonium, by decreasing the sediment’s adsorption capacities for these nutrients through physical and chemical processes [16,17]. Changes in salinity also profoundly affect the chemodiversity, composition, and bioavailability of organic matters in lake sediments [14]. These changes, along with shifts in the activities of the extracellular enzymes involved in microbial acquisitions of carbon and nutrients [18,19,20], profoundly influence microbial metabolism and SOC turnover [21]. Thus, understanding microbial metabolic traits across different salinity levels is essential for a deeper understanding of SOC dynamics in lake sediments.

Microbes secrete the extracellular enzymes associated with C, N, and P acquisitions to catalyze the breakdown of polymeric organic matter into monomer molecules. This process is commonly regarded as the “rate-limiting” step for SOC decomposition. The amount of these extracellular enzymes that microbes synthesize and excrete reflects their metabolic demands [21,22,23,24]. Sinsabaugh et al. (2009) revealed that the key extracellular enzymes catalyzing the terminal reactions in hydrolyzing principal C, N, and P sources had a mean activity ratio of approximately 1:1:1 in soils and sediments globally. Based on this finding, Moorhead et al. (2013, 2016) developed a vector analysis based on enzyme stoichiometric ratios to quantify microbial metabolic limitations (MMLs). This method quantifies the deviation of the enzyme activity ratio from the global average as an indicator of MML. Specifically, this method computes the angle and length of the vectors in the biplots visualizing C:N vs. C:P enzyme activity ratios to reflect the relative N vs. P limitation and relative C limitation [25,26]. In recent years, vector analysis has been extensively applied in the MML studies of soils [27,28,29,30] and sediments [31,32]. And significant correlations have been identified between variations in organic carbon and MMLs, exhibiting variable correlation directions across different environmental contexts [30,31,33,34]. Thus, the MMLs derived from enzyme activity ratios can serve as effective proxies for microbial metabolic traits in lake sediments, helping to elucidate how microorganisms relate to SOC dynamics under changing salinity conditions.

The Tibetan Plateau is home to the world’s largest number of high-altitude lakes with a wide range of salinity [35]. At the same time, the majority of lakes have experienced marked changes in salinity due to climate change-mediated lake expansion [3,36]. The wide range of salinity and the rapid change in lake salinity makes the Tibetan Plateau both an ideal and necessary study system for research on MML patterns and MML–OC relationships. Here, using sediment samples across 25 lakes spanning a salinity range of 0.13‰~31.06‰ on the Tibetan Plateau, we aimed to test two hypotheses: (1) high salinity increases MMLs, directly or indirectly via nutrients and microbial diversity, and (2) MMLs in sediments are negatively correlated with SOC content. We infer that MMLs induce microbial mining for deficient resources by breaking down SOC, which leads to its decomposition and loss.

To explore these hypotheses, we quantified MMLs in sediments through a vector analysis based on enzyme stoichiometric ratios, and examined their relationships with biotic and abiotic factors. In this study, we demonstrated that the microbial metabolism in sediment was primarily limited by C and N, with stronger limitations observed at higher salinity levels. Salinity and sediment pH were key factors affecting microbial metabolic limitations (MMLs), either directly or indirectly by influencing nutrient availability and microbial diversity. Furthermore, the content of sediment organic carbon (SOC) showed a negative correlation with microbial C limitation, a relationship that weakened when salinity and pH were accounted for. These findings indicate that increased microbial C limitation, driven by rising salinity and sediment pH, may play a critical role in accelerating the decomposition of sediment organic matter and the subsequent loss of SOC.

## 2. Materials and Methods

### 2.1. Study Area and Sampling

We investigated 25 lakes distributed across an altitude range of 4031~5008 m in July~September 2020 (Figure 1). These lakes are mostly above the treeline (the average altitude of the alpine treeline in the Himalayas, located on the southern edge of the Tibetan Plateau, is 3633 m [37]), and lake depths range between 3.8 and 55.7 m, with an average of 21.4 m. The vegetation in most areas consists mainly of alpine meadows or shrubs, with little to no trees around the lakes, or only sparse small plants. The sampled lakes included 9 freshwater lakes (salinity < 0.5‰) and 16 saline lakes. The saline lakes can be further categorized into 8 subsaline lakes (salinity = 0.5‰~3‰), 6 hyposaline lakes (salinity = 3‰~20‰), and 2 mesosaline lakes (salinity > 20‰) (Figure 1). For each lake, we selected 1~4 sampling sites (more sites in lakes with larger area) located towards the center of the lake, resulting in a total of 44 sediment samples. We avoided sites near the shore to avoid wave disturbance on the sediments. We arrived at the sampling sites by motorboat and used a global positioning system to record longitude, latitude, and altitude. At each sampling site, we collected 2 L of water from the upper 50 cm surface layer by a 5 L Schindler sampler and retrieved the surface sediment at the same location with a 6 cm diameter gravity core. We filtered 1 L of water through glass fiber filter membrane with a pore size of 0.7 µm (GF/F, Whatman) and stored the membranes in the dark at −20 °C until chlorophyll-*a* (Chl-*a*) analysis. We stored the remaining 1 L water samples at 4 °C until C, N, and P analyses. We carefully collected only the first centimeter of sediments from the sediment cores because the top surficial layer of sediments typically possesses higher microbial cell density, extracellular enzymatic activities (EEAs), and microbial activities [38,39,40]. We well homogenized each sediment sample and separated it into two subsamples. We then stored the two subsamples at −20 °C until analysis. One was used to analyze the composition and diversity of microbial community and to quantify the EEAs, while the other was used to determine the C, N, and P contents. Freezing is a recommended procedure for sediment preservation when the EEAs cannot be determined immediately [41]. All equipment, tools, and containers used during sediment sampling were sterilized in advance to avoid contamination. Moreover, we used disposable gloves, spoons, and blades for mixing and collecting samples, ensuring that all equipment and tools were carefully cleaned between sample collections. All sampling containers were pre-washed with distilled water and autoclaved before use. In addition, we monitored in situ environmental parameters at each sampling site. Specifically, water depth was obtained by a bathymeter; temperature and salinity in the water column were measured using a multiparameter water quality probe (YSI Incorporated, Yellow Springs, OH, USA).

### 2.2. C, N, and P Content Determinations

We freeze-dried the surface sediments before chemical and biological analyses. For sediment total nitrogen (TN) and total phosphorus (TP) measurements, we passed the freeze-dried subsamples through a 100-mesh sieve after grinding. We determined the sediment TN using the combustion method by an elemental analyzer (EA 3000, Euro Vector, Pavia, Italy). We analyzed the sediment TP following molybdenum, antimony, and scandium colorimetry [42]. For TN and TP in water samples, we used persulfate digestion to convert all nitrogen forms and organic phosphorus into nitrate and phosphate, respectively. Subsequently, we quantified the concentrations of nitrate and phosphate using colorimetric methods [42]. We passed the freeze-dried subsamples through a 10-mesh sieve and mixed them with carbon dioxide-free water at sediment-to-water ratios of 1:5 and 1:2.5 (g/mL). For the first ratio, the mixture was oscillated for 30 min, left for 30 min, filtered to obtain the supernatant, and then measured for conductivity using a conductivity electrode; for the second ratio, the mixture was oscillated for 2 min, left for 30 min, and measured for pH using a pH electrode [43]. Measuring the pH and conductivity using freeze-dried sediment samples ensured uniformity and comparability across all samples and avoided issues caused by variations in the timing of sample collection. The water salinity, sediment pH, and conductivity probes were calibrated with standard solutions to ensure accuracy and consistency among all samples.

We mixed the freeze-dried sediments with water (1:20 sediment/water ratio, g/mL) and filtered them through a 0.45 µm glass fiber membrane to extract the dissolved inorganic nitrogen (DIN, i.e., NH_4_^+^, NO_3_^−^, and NO_2_^−^), dissolved inorganic phosphorus (DIP, i.e., PO_4_^3−^) and DOC in surface sediments. We diluted the solution with ammonia-free water at a ratio of 1:10 and subsequently analyzed sediment DIN and DIP using colorimetric methods. NH_4_^+^ and PO_4_^3−^ were quantified following the Nessler’s reagent and molybdenum blue spectrophotometric method, respectively. NO_3_^−^ was determined directly at 275 and 220 nm light wavelengths, while NO_2_^−^ was determined by reacting the sample with Griess reagent to form a colored compound [42]. We used a TOC analyzer (ET1020A, O.I. Analytical, College Station, Texas, USA) to determine SOC and DOC.

Additionally, we also measured water Chl-*a*, which could be used to indicate nutrients and organic matter from algal biomass since algal biomass can support the C acquisition and growth of the microbial community [44]. Water Chl-*a* was measured spectrophotometrically after extraction in 90% acetone following standard method [45].

### 2.3. Bacterial and Fungal Community Analyses

We extracted the sediment genomic DNA using the DNeasy PowerSoil Kit (QIAGEN, Hilden, Germany) according to the manufacturer’s protocols. Meanwhile, three blank controls were performed following the same procedures as the sample treatments to control for potential laboratory contamination. For bacteria, we amplified the V4 region of the 16S ribosomal RNA gene using the universal primers [515F, 5′-GTGYCAGCMGCCGCGGTAA-3′ and 806R, and 5′-GGACTACNVGGGTWTCTAAT-3′] via the polymerase chain reaction. For fungi, we amplified the internal transcribed spacer 2 (ITS2) region of the nuclear ribosome using the universal primers [gITS7F, 5′-GTGARTCATCGARTCTTTG-3′ and ITS4R, and 5′-TCCTCCGCTTATTGATATGC-3′]. The PCR amplifications were carried out with the following steps: an initial denaturation at 94 °C for 5 min, followed by 24 cycles consisting of 30 s of denaturation at 94 °C, 30 s of annealing at 56 °C, and 20 s of extension at 72 °C, with a final extension at 72 °C for 5 min. To assess the presence of contamination and validate the amplification process, negative controls (no-template controls, NTCs) were included in each PCR set. The PCR products were then analyzed using 1.5% agarose gel electrophoresis, purified using AMPure XP beads (Beckman, Brea, CA, USA), and quantified with the Qubit dsDNA assay kit using a Qubit 2.0 Fluorometer (Thermo Fisher Scientific, Waltham, MA, USA). The negative controls (extraction blanks and NTCs) were analyzed to detect any potential contamination across the workflow, and no amplification or DNA was detected in these samples. We then normalized the barcoded PCR products of bacterial or fungal samples at equal molar concentration and sequenced them with 2 × 250 bp paired-end on the Illumina HiSeq sequencing platform (Illumina Inc., San Diego, CA, USA).

Finally, we processed the bacterial and fungal sequences on the Quantitative Insights into Microbial Ecology (QIIME, v2.0) pipeline [46]. Briefly, for bacteria, sequences were denoised to obtain amplicon sequence variants (ASVs) following the processes of removing low-quality and redundant sequences and splicing sequences and removing chimeras with the “DADA2” package [47]. The specific taxonomic classifier was trained and generated based on the primers used for amplification, the length of our sequence reads, and the preformatted reference sequence and taxonomy for the SILVA ribosomal RNA database using the “q2-feature-classifier” algorithm [48]. Then, the taxonomy assignment of ASVs was performed based on the specific taxonomic classifier using the “classify-sklearn” algorithm [49]. For fungi, the sequence processing was the same as for bacteria except that the taxonomy annotation was performed based on the UNITE database [50]. Finally, both the bacterial and fungal ASVs were rarefied to the minimum sequence number to prevent variations in abundance or sampling intensity from interfering with the biodiversity calculations.

### 2.4. EEAs Assays

The enzymes we studied were β-1,4-Glucosidase (BG), cellobiohydrolase (CBH), β-1,4-N-acetylglucosaminidase (NAG), leucine aminopeptidase (LAP), and alkaline phosphatase (AP). These extracellular enzymes catalyze the terminal reactions in substrate hydrolysis to acquire C, N, and P. Specifically, BG and CBH catalyze cellulose degradation, NAG and LAP catalyze chitin and polypeptide degradation, AP catalyzes phospholipid and phosphosaccharide degradations. We measured the EEAs of these five enzymes using a fluorimetric microplate enzyme assay [51]. This method directly measures the fluorescence intensity of products released from the fluorimetrically labelled substrates added to sediment suspensions, eliminating the need for prior extraction and purification of enzymes. This approach allows for the rapid analysis of a large number of sediment samples and enzymes. We used substrates conjugated with highly fluorescent compounds, 7-amino-4-methyl coumarin (7-AMC) and 4-methylumbelliferone (4-MUB), to measure the activities of LAP and four other enzymes, respectively. The water used for incubation was collected in situ for each sample, filtered through 0.2-µm nylon filters to eliminate bacteria and particles, autoclaved to eliminate enzymatic activity, and then kept frozen until it acclimated to the incubation temperature for EEAs measurements. Additionally, sediment turbidity and phenolics affect the fluorescence intensity of 4-MUB and 7-AMC by causing quenching [52]. The extent of quenching differs across various sediment suspensions, making it necessary to apply a correction procedure for each sediment analyzed. This procedure involves using a specific standard in the same sediment matrix. Consequently, each plate included at least three replicates for each substrate (sample + buffer + substrate), a quenched standard (sample + buffer + 4-MUB/7-AMC), a substrate control (water + buffer + substrate), and an optional abiotic control (autoclaved sediment + buffer + substrate). Incubation was performed using 1 g of sediment samples, 10 mL of previously autoclaved water, and the corresponding substrate for each EEA with a final concentration standardized of 0.2 mM. This concentration was determined based on preliminary experiments, in which different gradient concentrations of the substrate were tested on the same sample to optimize the enzyme reaction. The buffer selection was based on the specific enzyme being tested. BG, CBH, and NAG were measured using 0.1 M MES buffer (2-[N-Morpholino]ethanesulfonic acid, pH 6.1), AP was assayed in 0.1 M MES buffer at pH 10, and LAP was tested in 0.05 M Trizma buffer at pH 7.8. After a 30 min incubation in the dark on a shaking table at an average ambient temperature of 15 °C (ranging from 12 °C to 19 °C during sampling), the reaction was stopped using sodium hydroxide solution (1 M) before measuring the fluorescence, then the fluorescence emission from the AMC or MUB product was quantified using a computerized microplate fluorimeter (SH1M-SN, Agilent BioTek), with excitation/emission wavelengths set to 364/445 or 365/455. The EEAs were expressed as nanomoles of AMC or MUB released per gram of dry sediment per hour (nmol g sediment ^−1^ h^−1^).

### 2.5. MMLs Calculations

To quantify MMLs, we applied the vector analysis method based on the global ecoenzymatic stoichiometry law for C:N:P EEAs near 1:1:1 [21,26]. The quantified values indicate the relative elemental demands of the microbial community regardless of the variations in total EEAs. Vector analysis was conducted by the following equations [26].(1)$$\mathrm{Microbial}\;C\;\mathrm{limitation}=\mathrm{Vector}\;\mathrm{length}=\sqrt{x^{2}+y^{2}}$$Microbial N/P limitation = Vector angle (°) = atan2 (*x*, *y*) × 180/π(2)
where x is the relative activity of C- vs. P-acquiring enzymes; y is the relative activity of C- vs. N-acquiring enzymes. Microbial C limitation is quantified by the vector length, i.e., the square root of the sum of *x*^2^ and *y*^2^ (Equation (1)). Longer vector length corresponds to higher microbial C limitation. Specifically, long vector means that enzymes associated with C acquisition has relatively higher activity. This indicates that microbes excrete more enzymes to acquire carbon, which usually occurs when there is carbon limitation in the environment. Microbial N/P limitation is determined by the vector angle, i.e., the arctangent of the line extending from the plot origin to point (*x*, *y*) (Equation (2)). Vector angles above and below 45° indicate microbial P limitation and N limitation, respectively. For microbial P limitation, larger vector angle implies stronger limitation; while for N limitation, the trend is inverse.

### 2.6. Statistical Analysis

Our analyses intended to explore (1) the variations in MMLs along lake salinity, and the impacts of nutrients and microbial diversity on these variations, and (2) the correlations between MMLs and SOC. To achieve goal (1), we explored the following three aspects: the variations in MMLs, nutrients and EEAs across lake salinity gradients; potential drivers influencing microbial diversity; and the impacts and respective contributions of salinity, nutrients, and microbial diversity to MMLs.

For the first aspect, we used the linear regression to analyze the relationships of lake salinity with MMLs, sediment nutrients (i.e., SOC, TN, TP, SOC:TN, and SOC:TP), EEAs, and enzymatic stoichiometry (i.e., the C, N, or P acquisition EEAs and their proportions to total EEAs, the stoichiometry ratios between C, N, and P acquisition EEAs). All variables were log-transformed prior to these analyses in order to satisfy the assumptions of linear regression, such as normality and homoscedasticity. We further used analysis of variance (ANOVA) to compare MMLs and other environmental factors between freshwater, subsaline, hyposaline, and mesosaline lakes. Since sediment pH was suggested to be a key driver of MMLs, we also performed linear regression to analyze the relationships of sediment pH with MMLs, sediment nutrients, EEAs, and enzymatic stoichiometry.

For the second aspect, we quantified the relative contributions of environmental factors to microbial species richness using random forest [53]. We then examined the significance and predictive power of geographic distance and multiple environmental factors in explaining bacterial and fungal community composition differences using multiple regression on dissimilarity matrix (MRM) approach. This approach allowed us to identify linear, nonlinear, or nonparametric relationships between a community dissimilarity matrix and any number of explanatory distance matrices [54].

For the third aspect, we quantified the relative contributions of each environmental factor and microbial diversity to MMLs using random forest analysis. The microbial diversity included the richness of bacteria and fungi and the community structure of bacteria and fungi represented by the first axis of detrended correspondence analysis of the communities (DCA1). In addition, we partitioned the explanatory variables into the following five main driver categories: climatic factors, geographical features, pH and salinity, nutrients, and microbial diversity. Climate factors included the annual mean temperature (MAT) and annual mean precipitation (MAP), which were downloaded from the CHELSA dataset (Climatologies at high resolution for the Earth’s land surface areas, https://chelsa-climate.org) [55,56]. These data were then extracted based on the longitude and latitude of the sampling sites using the R package “raster V3.6-31”. The geographical feature was the elevation, with the geographic distance calculated from the longitude and latitude as an additional geographical feature input to the MRM analyses of bacterial and fungal community structures. Nutrients included TN, TP, TN:TP ratio, and Chl-*a* in water and TN, TP, SOC, NH_4_^+^, NO_3_^−^, NO_2_^−^, DIP, DOC, SOC:TN ratio, SOC:TP ratio, and TN:TP ratio in sediments. Since no significant effects of climatic and geographical factors on microbial C or N/P limitations were found according to the random forest analyses, these two abiotic factors were excluded from the following analyses. For the remaining three driver categories, we further quantified their pure/shared effects and direct/indirect effects on MMLs by performing variance partitioning analysis (VPA) and partial least squares path modeling (PLS-PM), respectively. VPA partitioned the variation in MMLs into components explained by the above three categories and their shared effects [57]. PLS-PM offered a framework to analyze the interrelationships among a set of variable blocks, including MMLs and the above three categories, taking into account prior knowledge of the phenomenon being studied [58]. In the PLS-PM, we performed an ANOVA to estimate the significance of the standardized path coefficient (β) and the model. The standardized total effect of each driver category on MMLs was then calculated from β values, encompassing both direct and indirect correlations.

To achieve goal (2), we used the linear regressions to analyze the correlations between MMLs and SOC. We further fitted the partial linear regressions between MMLs and SOC by controlling for water salinity or sediment pH in order to disentangle the potential control of water salinity or sediment pH on the correlations between MMLs and SOC. Before performing the random forest, VPA, and PLS-PM analyses, potential predictors were selected based on the high variance inflation factors (>10) to avoid strong collinearity or multicollinearity, using the R package “usdm V1.1–18” [59]. The predictors were Z score-transformed to allow comparisons in the MRM and PLS-PM analyses. The OLS regression, random forest, MRM, VPA, and PLS-PM analyses were conducted with R packages “car V3.1” [60], “randomForestSRC V3.2.2 [53]”, “ecodist V2.0.9 [61]”, “vegan V2.6” [57], and “plspm V0.5.0” [58], respectively.

## 3. Results

### 3.1. Nutrients, EEAs, and MMLs

For nutrients, in the water column, the mean values of TN and TP were 1.05 and 0.368 mg L^−1^, respectively (Appendix A), and both increased significantly with water salinity (*p* < 0.05, Appendix A). TN:TP decreased significantly from freshwater to saline lakes (*p* < 0.01, Appendix A). The Chl-*a* concentration was generally low with a mean value of 1.33 µg L^−1^ (Appendix A) and had no obvious differences among salinity levels (*p* > 0.05, Appendix A). In the surface sediment, the mean values of SOC, TN, and TP were 42.7, 4.81, and 0.56 g kg^−1^, respectively (Appendix A). As water salinity increased, SOC, SOC:TN, SOC:TP, and TN:TP decreased significantly (*p* < 0.05, Figure 2a,d–f); NH_4_^+^, NO_3_^−^, NO_2_^−^, and PO_4_^3−^ increased significantly (*p* < 0.05, Appendix A); and TN, TP, and DOC showed no obvious changes (*p* > 0.05, Figure 2b,c and Appendix A).

For EEAs, as the salinity increased, the activities of the C- and N-acquiring enzymes increased (*p* < 0.01, Appendix A) and the P-acquiring enzyme decreased (*p* < 0.01, Appendix A). Consequently, the activity ratios of C- to P-acquiring and N- to P-acquiring enzymes (enzyme C:P ratio and N:P ratio) also increased markedly (*p* < 0.01, Appendix A). For MMLs, the enzyme C:N ratio and C:P ratio indicated that the sediment microbial community was more limited by C sources than by N or P sources (Figure 3a,b). The vector analysis further reflected that N sources were more limited than P sources for microbes since most of the vector angles were below 45° (Figure 3c). As salinity increased, vector length, indicating C limitation, significantly increased (*p* < 0.001, Figure 3e), particularly during the transition from freshwater to subsaline lakes (*p* < 0.05, Figure 3d). The vector angle exhibited a significant linear negative correlation with water salinity (*p* < 0.001, Figure 3g), indicating a substantial increase in microbial N limitation as water salinity increased. Water salinity accounted for 23% and 44% of the variations in microbial C and N/P limitation, respectively (*p* < 0.001, Figure 3e,g).

Furthermore, C, N, and P contents, EEAs, and MMLS and their relationships found in our study are broadly similar to previous work on saline lakes and freshwater lakes (Appendix A). For EEAs and enzymatic stoichiometry, the proportion of P acquisition EEA was lower in saline lakes than in fresh waters (*p* < 0.05, Appendix A). Microbial C limitation was generally higher in saline lakes than in freshwater lakes, with the average vector length being significantly higher in the former (1.125 and 0.933 in Qinghai Lake and Hulun Lake, respectively) than in the latter (0.678 and 0.835 in Fuxian Lake and lakes along the Yangtze–Huaihe River basin, respectively) (*p* < 0.001, Appendix A). For microbial N/P limitation, saline lakes were N-limited at the majority of sites with an average vector angle below 45° (42.07°, 33.15°, and 39.40° in Qinghai Lake, Hulun Lake, and Tibetan lakes, respectively), while freshwater lakes demonstrated a conversion between N and P limitation with an average vector angle close to 45° (45.47° and 45.06° in Fuxian Lake and lakes along the Yangtze–Huaihe River basin, respectively) (Appendix A, Appendix A).

### 3.2. Bacterial and Fungal Diversities and Their Driving Factors

We obtained an average of 122,967 (103,703–135,642) reads for bacteria and 14,939 (13,195–16,753) reads for fungi after denoising. Thus, the bacterial and fungal ASVs were finally rarefied to the minimum sequence numbers of 103,703 and 13,195, respectively. The rarefaction curves for both bacteria and fungi reached a plateau, confirming that the sequencing effort was sufficient to represent the microbial diversity in the samples (Appendix A). A total of 78 phyla and 609 orders of bacteria and 15 phyla and 53 orders of fungi were identified across the sediments of the Tibetan Lakes. The most dominant bacterial phyla were *Proteobacteria*, *Bacteroidetes*, and *Firmicutes*, with mean relative abundances of 26.98%, 14.72%, and 12.79%, respectively, followed by *Chloroflexi* (11.14%), *Actinobacteria* (8.45%), and *Cyanobacteria* (4.85%) (Appendix A). The top two fungal phyla identified were *Ascomycota* (25.13%) and *Basidiomycota* (8.92%), followed by *Rozellomycota* (2.49%) and *Chytridiomycota* (1.88%) (Appendix A).

With regard to both species richness and community structure, the bacterial community was dominantly impacted by sediment pH and water salinity (*p* < 0.01, Figure 4a,c), and the fungal community was impacted by water salinity, sediment pH, and water TN:TP (*p* < 0.05, Figure 4b,d). In addition, bacterial richness was secondarily mediated by nutrients, including water TP and TN:TP and sediment DIP (*p* < 0.05, Figure 4a), and the bacterial community structure was secondarily mediated by sediment TN and TP and geographic distance (*p* < 0.05, Figure 4c). Fungal richness was additionally affected by water TP and sediment SOC:TN (*p* < 0.05, Figure 4b), and the fungal community structure was affected by water TN and geographic distance (*p* < 0.01, Figure 4d).

### 3.3. Drivers of MMLs and Correlations Between MMLs and SOC

For the relative contributions of individual factors, sediment pH exhibited the greatest effects on both microbial C and N/P limitations (*p* < 0.01, Figure 5a,d). Additionally, the DCA1 of bacteria, TN, and sediment DOC also made significant contributions to the variation in microbial C limitation (*p* < 0.05, Figure 5a), whereas bacterial richness, the DCA1 of fungi, and water TN showed relatively greater contributions to the variation in microbial N/P limitation than other factors (*p* < 0.01, Figure 5d). Climatic and geographic factors, including MAT, MAP, and elevation, showed nonsignificant impacts on MMLs (*p* > 0.05, Figure 5a,d).

For the pure and shared effects of pH and salinity, nutrients, and microbial diversity on MMLs, pH and salinity explained 5% of the microbial C limitation (Figure 5b), while nutrients accounted for the highest pure effect of 15% for microbial N/P limitation (Figure 5e). The shared effects between pH and salinity and microbial diversity consistently showed the highest contributions among all shared effects across the three categories, accounting for 30% of the microbial C and 40% of the microbial N/P limitation (Figure 5b,e).

For the direct and indirect effects of pH and salinity, nutrients, and microbial diversity on MMLs, pH and salinity demonstrated the maximum direct influences, with path coefficients of 0.82 for microbial C limitation and −0.42 for microbial N/P limitation, and pH and salinity and nutrients both showed indirect influences on microbial C and N/P limitations via microbial diversity (Figure 5c,f). Consistent with our previous results (Figure 4), pH and salinity had a decisive advantage over nutrients in regulating microbial diversity in the PLS-PM model (Figure 5c,f). In terms of the total effect of each of the three categories revealed by VPAs and PLS-PM, pH and salinity consistently represented the greatest influences for both microbial C and N/P limitations (Figure 5b,c,e,f).

Furthermore, we also found significant correlations of sediment pH with MMLs, sediment nutrients, EEAs, and enzyme stoichiometry. Sediment pH exhibited significant linear negative relationships with microbial N/P limitation and AP enzyme activity (*p* < 0.001, Appendix A), as well as positive relationships with microbial C limitation and the ratios of enzyme C:P and N:P (*p* < 0.001, Appendix A). The SOC, SOC:TN, and SOC:TP decreased significantly with increasing sediment pH (*p* < 0.05, Appendix A).

According to linear regressions, the SOC showed a significant negative correlation with microbial C limitation (*p* < 0.01, Figure 6a,d), but showed no significant correlation with microbial N/P limitation (*p* > 0.05, Figure 6b,e). After controlling for the water salinity, microbial C limitation decreased its effect on the variation of SOC from 21% to only 9% (*p* < 0.05, Figure 6c). After controlling for the sediment pH, microbial C limitation showed a nonsignificant relationship with SOC (*p* > 0.05, Figure 6f). These analyses showed that fluctuations in water salinity and sediment pH could, respectively, increase and recouple the linkage between MMLs and SOC.

## 4. Discussion

### 4.1. C and N Limitations Dominated MMLs in Tibetan Lakes

In the Tibetan Plateau, microorganisms in lake sediments were primarily limited by C and N based on the vector analyses of enzymatic stoichiometry (Figure 3a–c). This is consistent with previous reports on single saline lakes [20], marine gulfs [62], saline wetlands, and soils [63,64] as well as freshwater rivers and streams [65], but inconsistent with the results of freshwater lakes (Appendix A, W. Zhang unpubl.) and wetlands [66]. It is worth noting that although nitrogen limitation may occur intermittently in many ecosystems, recent evidence suggests that nitrogen saturation, rather than limitation, is increasingly observed in ecosystems, especially in aquatic ecosystems, such as streams [67,68,69]. In saline lakes, we found that N limitation was more dominant compared to P limitation, particularly at higher salinity levels (Figure 3g). In contrast, freshwater lake sites are roughly equally divided between N and P limitations (Appendix A). This was consistent with the observation that the water TN:TP ratio decreased with salinity (Figure 2f and Appendix A), and that sediment P availability (i.e., PO_4_^3−^ concentration) increased with salinity (Appendix A). Mechanistically, as salinity increases, sediment’s ability to adsorb phosphorus decreases [17], increasing phosphorus availability for microbes. The elevated PO_4_^3−^ concentration reduced the microbial secretion of AP, thereby lowering AP activity, while simultaneously inducing N limitation due to stoichiometric imbalance, thus promoting N acquisition EEAs for the cellular maintenance of microbes [70]. This commonly occurs in saline waters and soils [71,72]. Consistently, our results showed that P-acquiring EEAs decreased while N-acquiring EEAs increased significantly with increasing salinity (Appendix A).

### 4.2. Predominant Contributions of Water Salinity and Sediment pH to MMLs

Our findings support hypothesis (1), that increasing water salinity would increase MMLs, directly or indirectly via nutrients and microbial diversity (Figure 5c,f). Specifically, for the first time, through multi-lake surveys conducted along salinity gradients, we demonstrated higher degrees of microbial C and N limitations as salinity increased. This finding was not entirely consistent with the case studies reported earlier. For example, in the sediments of the saline Qinghai Lake, microbial C limitation increased while N limitation showed no significant change with increasing salinity [20]. In the sediments of a mangrove wetland, salinization exacerbated microbial N limitation while relieved C limitation [64]. These differences suggest that the response of MMLs to salinity may be influenced by factors such as habitat and the range of salinity values studied, and that the response mechanisms cannot be simply extrapolated from one study to another. In this study, for instance, it is hard to squarely place the mechanisms of salinity since some potential confounding factors, such as sediment pH, nutrient availability, and microbial diversity, changed as salinity increased. These factors may play key roles in regulating MMLs (for further explanations, see below). Therefore, further studies are needed to validate these mechanisms before extrapolating them to other sites or systems.

Sediment pH also substantially altered EEAs and MMLs in lake sediments, with high contributions of up to 44% and 50% on microbial C and N/P limitations, respectively. This might be naturally attributed to the fact that pH was the dominant control on rates of enzyme-catalyzed reactions through affecting the ionization groups of the enzyme molecules or influencing the ionization state of the substrate, or both [73]. Additionally, under alkaline conditions, as the pH increases, the release of phosphorus bound to iron, aluminum, and other oxides in the sediments also increases. This leads to an enhancement of bioavailable phosphorus at the sediment–water interface [74], which helps alleviate phosphorus limitation for sediment microorganisms. This observation is consistent with our study, where we found that alkaline conditions resulted in decreased AP activity as sediment pH increased (Appendix A). pH has been shown to be the most important driver of EEAs in various systems, including the Songnen Plain and Tibetan alpine meadow in China [75,76], temperate grasslands and croplands across Britain [77], and acidic tundra heaths in northwestern Finland [78]. Furthermore, pH can induce differential effects on the reaction rates of the decompositions of C, N, and P resources catalyzed by extracellular enzymes, ultimately influencing the relative availability of these elements to microbes, which is reflected in changes in MMLs [79].

Additionally, salinity and pH could affect MMLs indirectly via microbial diversity. This hypothesis is plausible because of the dominant controls of salinity and pH on microbial diversity and the close links between microbial diversity and MMLs. Unsurprisingly, the linkages of microbial diversity with salinity or pH have been reported in past studies. A sediment investigation across nine saline lakes on the northern Qinghai–Tibetan Plateau found that salinity, instead of geographic distance, shaped bacterial alpha diversity and community structure [80]. A review of Australian saline lakes identified salinity as the primary factor determining planktonic biodiversity [81]. As observed in this study, an increase in microbial diversity does not always correlate with resource abundance; instead, it may result in more severe resource limitations due to factors such as resource competition [82] and uneven resource utilization [83]. As microbial diversity increases, the greater variety of species within the community intensifies competition for resources [84]. These microorganisms may compete for the same substrates or resources (such as nutrients and carbon sources), making certain resources scarcer and heightening the potential for resource limitations [82]. In communities with high diversity, different microorganisms may have varying efficiencies and rates of resource utilization; some microorganisms may occupy more advantageous ecological niches, which restricts other microorganisms in their access to resources [85]. This uneven utilization can lead to the overall depletion of resources, increasing the degree of resource limitation [83].

Notably, the impacts of salinity and pH on MMLs were found to surpass those of nutrients, climate, and microbial diversity in the lake sediments of the Tibetan Plateau. In contrast, in the largest saline lake in China, bacterial community structure emerged as the primary determinant influencing the dynamics related to microbial C and N limitations, with salinity or nutrient levels following as secondary factors [20]. These discrepancies implied the scale dependency of disentangling the dominant drivers of MMLs along salinity gradients. Our multi-lake study covered a broader range of salinity, allowing us to identify the crucial role of salinity gradients in the variations of MMLs. In contrast, single-lake studies are limited to narrower salinity ranges, where sampling effects tend to emphasize the driving variables for MMLs that exhibit small-scale specificity. Given the marked changes in the salinity of inland waters, induced by intensifying climate warming [3], our findings underscore the necessity of studying MMLs to comprehend how biogeochemical cycles in lake sediments vary within salinization or desalinization processes.

### 4.3. SOC Is Negatively Associated with Microbial C Limitation

The observed significant negative association between microbial C limitation and SOC is consistent with hypothesis (2). We also found that the association between SOC and carbon limitation weakened or disappeared statistically when controlling for salinity or pH (Figure 6a,c,d,f). This meant that, compared to situations without salinity and pH disturbances, the same reduction in SOC in the presence of these disturbances would subject microbes to a stronger C limitation, or an increase in microbial C limitation would lead to greater carbon loss. Our study offers new insights into the potential mechanisms that may generate negative relationships between SOC and water salinity or sediment pH (Figure 2a and Appendix A). Conventionally, an increase in SOC is thought to relieve microbial C limitation. However, an alternative explanation may also hold true. Increased salinity or pH could somehow change the physiology of microbes so that they are less efficient in using carbon [14,15,86] and thus need to synthesize more C-acquiring enzymes, manifesting as higher values of microbial C limitation, thereby driving the decomposition of sediment organic matter and subsequent SOC loss.

Specifically, potential explanations for why carbon becomes more limited for microbes at higher salinity could be the reduced carbon availability, side effects of resistance strategies, and the weakening of the carbon utilization capacity. As water salinity increases, the content of refractory OC (e.g., lignin) increases, leading to reduced carbon availability for microbes [14,87]. From the microbial perspective, they may acclimate to salinity stress through a series of resistance mechanisms, such as accumulating organic compatible solutes to sustain osmotic pressure [11], and these strategies come at the cost of large investments in carbon and energy [86,88]. Moreover, increasing salinity has been documented to reduce the carbon utilization ability of certain dominant sediment microbes without altering their carbon utilization preference [15], potentially making carbon more limited to microbes. These factors might collectively stimulate microbial investment in labile OC acquisition in lakes with higher salinity. This is supported by our findings that saline lakes had either higher EEAs or higher relative investments in labile C-acquiring enzymes, including BG and CBH (Appendix A). These findings are consistent with previous lake sediment incubation experiments showing that higher salinity significantly enhanced the mineralization (or loss) of OC after adding fresh organic matter, known as positive priming effects [89,90]. These positive priming effects could be partially attributed to the microbial C limitation and more C-acquiring enzyme investment induced by rising salinity, as per the findings of our study.

As discussed above, the significant negative association between C limitation and SOC content may be driven by an interactive mechanism between SOC and C limitation, rather than a unidirectional effect. This mechanism underscores how microbial-mediated changes in SOC storage respond to and influence variations in microbial C limitation under altered water salinity or sediment pH in lake ecosystems. Consistently, soil salinity has been found to have a negative correlation with soil OC content, as indicated by meta-analyses of soil samples worldwide [91,92]. Specifically, an increase in soil salinity from 1 to 5 dS m^−1^ in global croplands and noncroplands would be associated with soil OC losses of 6.97% and 16.12%, respectively [91]. Additionally, pH also has been found to be significantly negatively correlated with soil OC in global grasslands, as demonstrated in a meta-analysis of 257 studies [93]. Our findings may provide an explanation for these relationships from the perspective of microbial metabolic traits.

Together with previous studies examining the connections between MMLs and OC predominantly focusing on soils rather than sediments, we found a recurring trend: OC loss tended to coincide with environmental stressors detrimental to indigenous microorganisms. This trend largely reflected shifts in MMLs due to deteriorating environmental conditions, including elevated pH or salinity (this study), heavy metal contamination, reduced precipitation or intensified drought, and the adoption of plastic film mulching [30,33,34]. For instance, the heavy metal contamination of soils aggravated microbial C limitation in a zinc smelter in northwest China, directly increasing C release by promoting soil OC decomposition [33]. In contrast, in an arid and semi-arid grassland region located in the Chinese Loess Plateau, reduced precipitation led to decreased vegetation and, consequently, lower soil organic matter. It also indirectly led to the loss of soil organic matter by exacerbating microbial C and P limitations [34]. Additionally, under long-term plastic film mulching, microbial P limitation was influenced by soil temperature, moisture, and pH, and was indirectly exacerbated by the loss of soil OC, as the decomposition of soil OC is a key source of soil-available phosphorus [30]. Therefore, higher attention should be directed towards investigating MML–OC connections to unravel the underlying mechanisms driving ecosystem OC dynamics.

### 4.4. Study Limitations and Future Directions

This study has several important limitations that should be addressed in future research. Firstly, the temporal variability of salinity. While this study examined changes in MMLs with salinity by analyzing lakes across a broad range of salinity, it does not fully capture how MMLs change over time at the same location as water salinity fluctuates. For example, with the increasing warming and wetting of the Tibetan Plateau, lake water salinity has been decreasing over the past few decades [3]. The true trend of MML changes during this period should be continuously monitored. Secondly, potential unconsidered confounding factors, such as seasonal differences and the composition of sediment organic carbon. This study only collected a single summer sample at each site, leaving it unclear whether seasonal variations in temperature and other factors might influence the overall trend of MML changes with salinity. For example, a recent study reported that the significant decline in lake OC burial throughout the Holocene was closely linked to changes in temperature seasonality, based on records from lake cores across the Tibetan Plateau [94]. In addition, the composition of SOC may change with salinity [14], affecting microbial enzymatic investments and organic carbon decomposition. How this factor influences the MML–SOC relationship remains uncertain. Addressing these issues will deepen our understanding of the changes in lake organic carbon pools and their microbial mechanisms. Finally, our results suggest that changes in salinity may alter the OC content in lake sediments by affecting microbial metabolic traits. This could offer new insights for improving prediction models of lake SOC stocks. Existing models in this field often give limited attention to the microbial mechanisms driving SOC changes (e.g., the microbial allocation of extracellular enzymes in response to resource imbalance) and rarely explore the potential impact of salinity fluctuations on SOC dynamics [95,96,97]. However, the salinity in global lakes has indeed undergone clear changes in recent decades [1,2,3]. Incorporating and refining these aspects could enhance the accuracy of dynamic models for lake SOC stocks.

## 5. Conclusions

Collectively, we clarified that sediment microorganisms in Tibetan lakes primarily experienced C and N limitations, which intensified with higher water salinity. Furthermore, we found that microbial C and N limitations were predominantly changed by water salinity and sediment pH, followed by microbial diversity and nutrients. Due to the significant negative correlation between microbial C limitation and SOC, this study suggested that intensified microbial carbon limitation, induced by increasing water salinity and sediment pH, could be a crucial mechanism leading to the increased decomposition of sediment organic matter and subsequent loss of SOC. Given the marked changes in the salinity of inland lakes due to global climate warming, our findings may prove vital in the research of C cycles in aquatic ecosystems, such as updating the existing prediction models of lake SOC stocks by taking salinity effects into account.

## Figures and Tables

**Figure 1 microorganisms-13-00629-f001:**
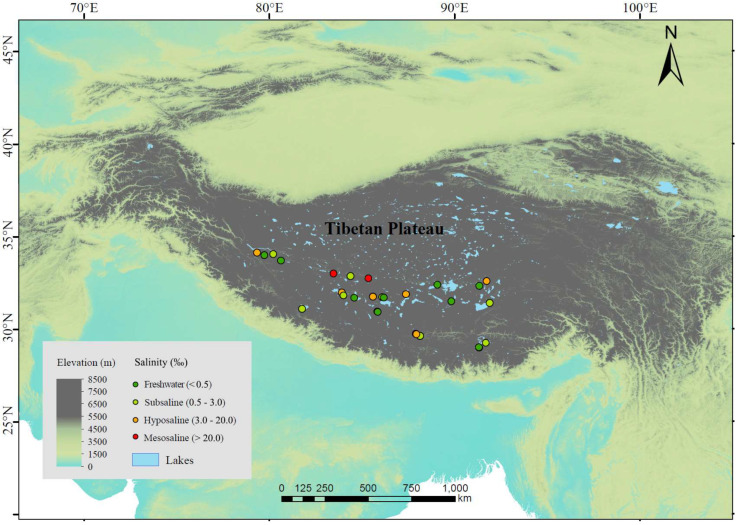
Sampling sites of lake sediments in Tibetan Plateau. Dark green, light green, orange, and red points represent freshwater (salinity < 0.5‰), subsaline (salinity = 0.5‰~3‰), hyposaline (salinity = 3‰~20‰), and mesosaline (salinity > 20‰), respectively. Elevation is also shown according to color gradients.

**Figure 2 microorganisms-13-00629-f002:**
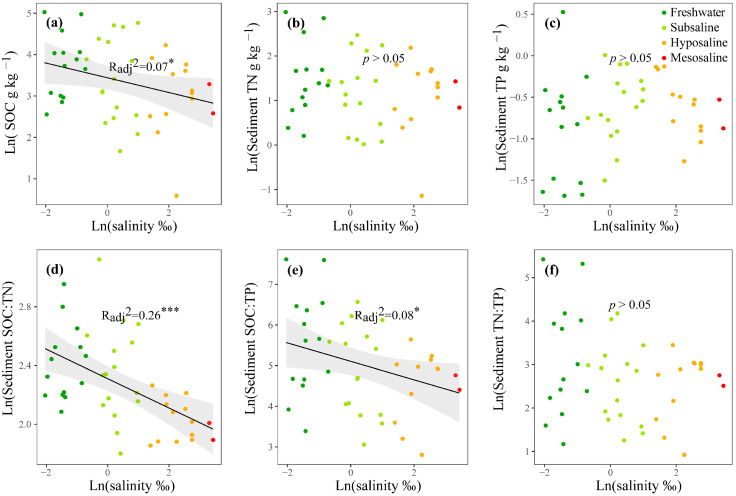
The linear regressions of the log-transformed water salinity with the SOC, TN, TP, SOC:TN, SOC:TP and TN:TP, and TP in the surface sediment (**a**–**f**). Only the significant fitted linear regressions are plotted with 95% confidence intervals filled in gray. Dark green, light green, orange, and red points represent freshwater (salinity < 0.5‰), subsaline (salinity = 0.5‰~3‰), hyposaline (salinity = 3‰~20‰), and mesosaline (salinity > 20‰), respectively. SOC: sediment organic carbon. TN: total nitrogen. TP: total phosphorus. The adjusted R^2^ values of the linear models are denoted. * *p* < 0.05, *** *p*< 0.001.

**Figure 3 microorganisms-13-00629-f003:**
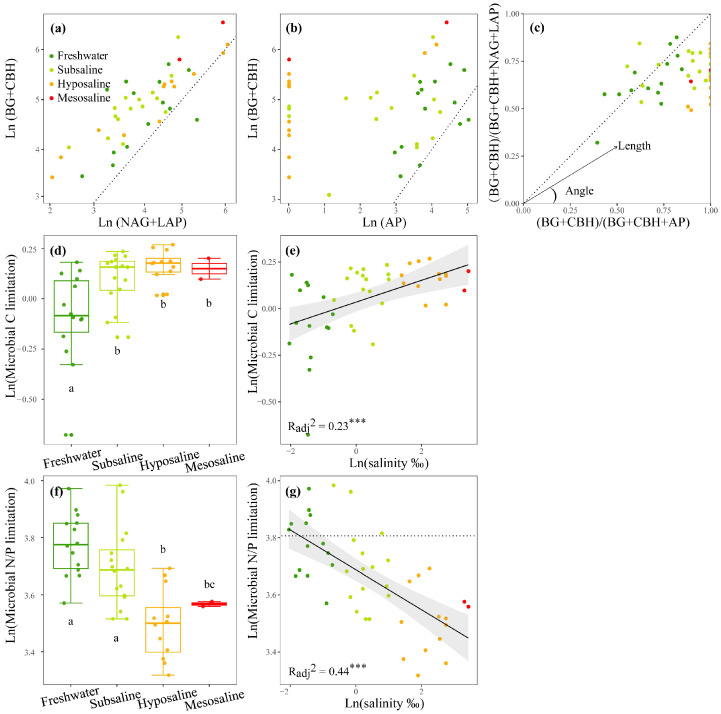
The stoichiometry of the enzymes C:(C+N) ratio (**a**) and enzymes C:(C+P) ratio (**b**), the vector analysis of enzymatic stoichiometry (**c**), a box plot comparing microbial C or N/P limitations among different salinity levels (**d**,**f**), and the linear correlations of logarithmically transformed lake salinity with logarithmically transformed microbial C limitation (**e**) and microbial N/P limitation (**g**) of lakes sediments in the Tibetan Plateau. BG, β-1,4-glucosidase; CBH, β-D-cellobiosidase; NAG, β-1,4-N-acetylglucosaminidase; LAP, L-leucine aminopeptidase; AP, alkaline phosphatase. A 1:1 dotted line is superimposed (**a**–**c**), which indicates equal investments into the compared element acquisition EEAs on the x and y axis. The letters above or below the boxes denote significances of differences in microbial C or N/P limitations among different salinity levels (**d**,**f**). A y = ln(45) dotted line is added (**g**) to estimate microbial N/P limitation, dots below (or above) are identified as N (or P) limitation. Dark green, light green, orange, and red points represent freshwater (salinity < 0.5‰), subsaline (salinity = 0.5‰~3‰), hyposaline (salinity = 3‰~20‰), and mesosaline (salinity > 20‰), respectively. The adjusted R^2^ values of the linear models are denoted. *** *p* < 0.001.

**Figure 4 microorganisms-13-00629-f004:**
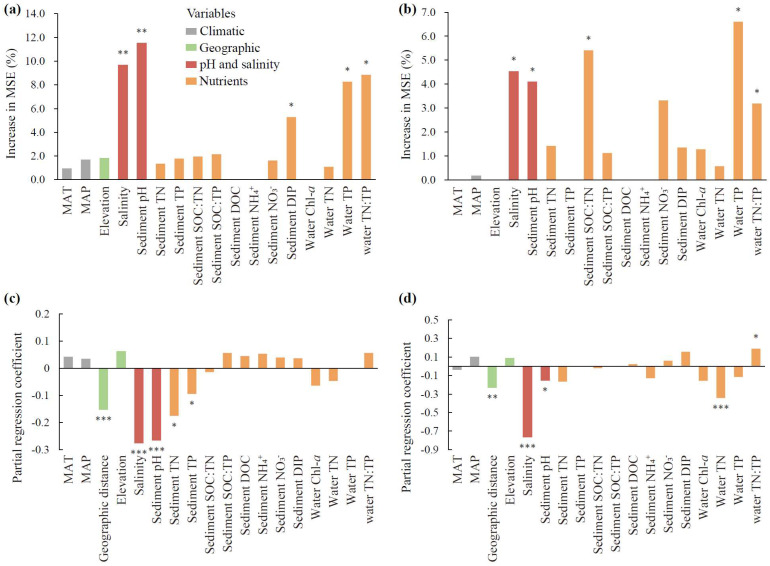
The contributions of abiotic factors on microbial community attributes. The panels include random forest analyses of the richness of bacteria (**a**) and fungi (**b**), and multiple regression on dissimilarity matrices of bacterial (**c**) and fungal (**d**) communities. The columns filled with grey, green, orange, and red indicate climatic, geographic, nutrient variables, and pH and salinity, respectively. MAT: mean annual temperature. MAP: mean annual precipitation. SOC: sediment organic carbon. TN: total nitrogen. TP: total phosphorus. DOC: dissolved organic carbon. NH_4_^+^: ammonia nitrogen. NO_3_^−^: nitrate nitrogen. DIP: dissolved inorganic phosphorus. Chl-*a*: chlorophyll-*a*. The significance of each variable is shown above the column. * *p* < 0.05, ** *p* < 0.01, *** *p* < 0.001.

**Figure 5 microorganisms-13-00629-f005:**
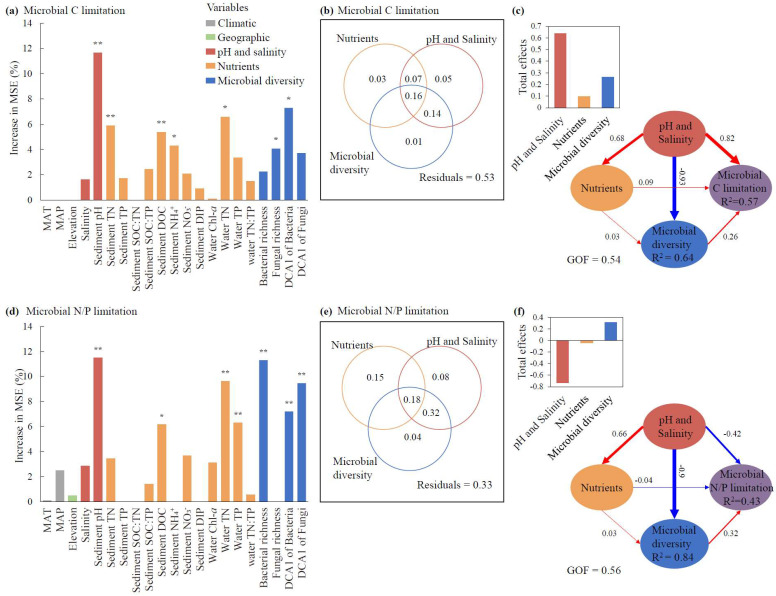
The effects of abiotic factors and microbial diversity on microbial metabolic limitations based on random forest analyses (**a**,**d**), variance partitioning analyses (**b**,**e**), and partial least squares path modeling (PLS-PM) (**c**,**f**). The total effects of nutrients, pH and salinity and microbial diversity on the microbial C and N/P limitations according to the PLS-PMs are shown in the top-left corners of panels (**c**) and (**f**), respectively. The path coefficients indicating the effect sizes are shown adjacent to the arrows with the arrow width proportional to the path coefficients and arrow color (red/blue) representing correlation directions (positive/negative), and the goodness of fit for PLS-PMs are denoted (**c**,**f**). The abiotic factors include climate factors, elevation, nutrients, and pH and salinity. Climate factors include MAT (mean annual temperature) and MAP (mean annual precipitation). Nutrients include Chl-*a*, TN, TP, and TN:TP ratio in water and TN, TP, SOC, NH_4_^+^, NO_3_^−^, DIP, DOC, SOC:TN, SOC:TP, and TN:TP in sediments. Microbial diversity includes the richness of bacteria and fungi, and the community structures of bacteria and fungi reflected by the DCA1 (first axis of detrended correspondence analysis of the community). The columns filled with grey, green, red, orange, and blue indicate climate factors, elevation, pH and salinity, nutrients, and microbial diversity, respectively. Since climate factors and elevation had nonsignificant effects on microbial C or N/P limitations (**a**,**d**), these two factors were excluded in the PLS-PM and variance partitioning analyses (**b**,**c**,**e**,**f**). For panels (**a**) and (**d**), the significance of each variable is shown above the column, * *p* < 0.05, ** *p* < 0.01. MAT: mean annual temperature. MAP: mean annual precipitation. SOC: sediment organic carbon. TN: total nitrogen. TP: total phosphorus. DOC: dissolved organic carbon. NH_4_^+^: ammonia nitrogen. NO_3_^−^: nitrate nitrogen. DIP: dissolved inorganic phosphorus. Chl-*a*: chlorophyll-*a*.

**Figure 6 microorganisms-13-00629-f006:**
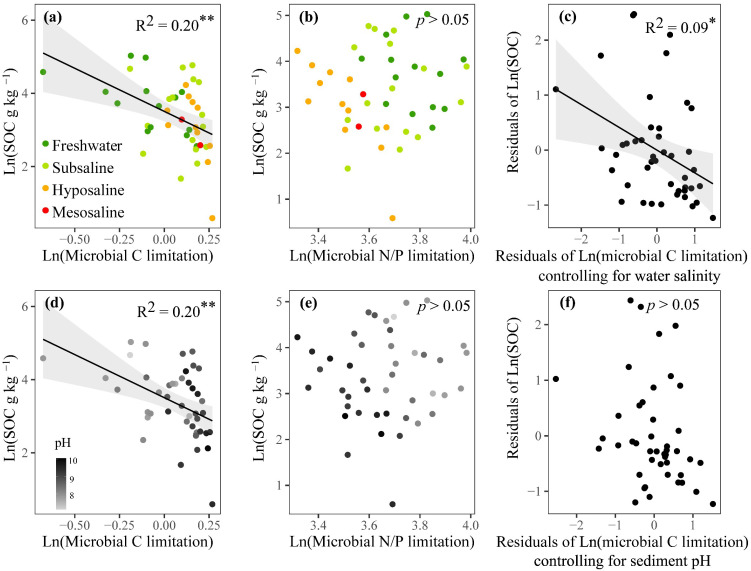
Relationships of SOC (sediment organic carbon) content with microbial resource limitations, including linear regressions between logarithmically transformed SOC and microbial C (**a**,**d**) or N/P limitation (**b**,**e**) and partial linear regressions between logarithmically transformed SOC and microbial C limitation when respectively controlling for water salinity (**c**) and sediment pH (**f**). Dark green, light green, orange, and red dots represent freshwater (salinity < 0.5‰), subsaline (salinity = 0.5‰~3‰), hyposaline (salinity = 3‰~20‰), and mesosaline (salinity > 20‰) sampling sites, respectively. For (**d**) and (**e**), dot colors represent pH values with deeper grey indicating higher pH. Only significant fitted linear regressions are plotted with 95% confidence intervals filled in gray. Adjusted R^2^ values of linear models are denoted. ** *p* < 0.01, * *p* < 0.05.

## Data Availability

The datasets supporting the conclusions of this article are included within the article and its additional file. The 16S rRNA and ITS2 sequences raw reads were deposited into the NCBI Sequence Read Archive (SRA) database under accession number PRJNA1194750 (https://www.ncbi.nlm.nih.gov/sra/PRJNA1194750, accessed on 6 December 2024).

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
