# Peer review of "Microbial Metabolic Limitations and Their Relationships with Sediment Organic Carbon Across Lake Salinity Gradient in Tibetan Plateau"

_microorganisms, 2025, doi:10.3390/microorganisms13030629_

Round 1

Reviewer 1 Report

Comments and Suggestions for Authors

In this paper, the authors use a gradient in salinity to test how microbial carbon and nutrient limitation/acquisition change across Tibetan lakes. In general, I thought that the question asked was interesting (although can be elaborated upon, see below), and the dataset itself seemed to be informative. Furthermore, the writing itself was quite good, and I feel like the topic should be of interest to other readers of Microorganisms. That said, I think there are also a few places that the authors can improve upon. Firstly, while the authors state that this salinization is an effect of climate change, it is not really clear how this is the case from the introduction. Not that I deny it is the case, i just think that it would be relevant information to communicate at the beginning of the paper. In the methods, I think that there could also be more description about the lakes themselves and how the samples were taken. Notably, there is nothing about lake depth, which to me is a glaring omission. There are also some important points about the microbial community characterization and EEA data generation that are missing. For example, there is little information on the number of sequences retrieved, the number per sample, what the rarefaction threshold was, etc. For the EEA, I think it is important to know the standards that were used, the concentration of the substrates, if the lake water was autoclaved etc. I think the readers may be interested in these details. Lastly, I would be careful in over attributing the results to salinity, as it seems there are a lot of things responsible for the results. Other mores specific comments follow:

line 19-20: not sure I fully understand this sentence about MML

line 37: maybe can expand on this first sentence a bit, since it seems to justify the entire project. How is climate warming leading to both salinization and desalinization? I think its even interesting if the point is to just test how things change over a salinity gradient, but if climate change is the root justification, then it must be established how climate change produces salinity changes

line 91: doesn’t have to be altered…..even if it’s a natural gradient, it still works well

line 94: hypothesis 2 is fairly weak in my opinion, given that you are basically just saying that there is an association between the two of these things. For example, I think it would make more sense if you hypothesized that the relationship should be positive or negative and then explain why you would think this

line 100: what was the average number of samples per lake?

line 100: also, how deep were the lakes, and how did they vary? Lake depth could be quite an important factor here….  also, will they reflect the water chemistry if its taken from the surface?

line 119: you mean the fraction that was frozen at -20?

line 129-130: are these standard methods?

line 176: Maybe it would be good to give the range of reads, the average/median number, and then the value to which the reads were actually rarefied to? Just to have an idea

line 181: Actually, what is in the watershed of these lakes…..are they mostly above treeline, or is there a lot of terrestrial vegetation? I guess my question is how much cellulose might we expect to be entering the lakes?

line 192: I think it’s a good idea to use water from the respective waterbodies to run the incubations. However, might it be possible that there are also enzymes in the water column, and that they might be different than the enzyme profile in the sediments? How might you account for this? Most studies are boiling/autoclaving the water before to deactivate the enzymes, such that only the sediment EEA is measured…..

line 194: what concentration were the substrates? Its important that they are saturating conditions…..were they kept on a shaking table while incubating?

line 198: did you stop the reaction with glycine buffer or similar before reading? Otherwise the fluorescence will keep increasing…..

line 200: did you run the samples with any controls (such as for quenching, background fluorescence, standards, etc)? Its just important for the readers to know how the analyses were done so that they can have faith in the data as they read on…..

line 229-230: what are the cutoffs between these categories of freshwater, subsaline, hyposaline, and mesosaline? How many samples did you have for each? After reading farther, I see that this information is in the results section. However, could think about moving it up? But I guess this is up to you

line 246: what is MAT and MAP?!? I don’t think you have talked about these two acronyms before… Will be necessary to state how these data were collected, compiled, etc, and acronym must be spelled out upon first use!

line 250: I also don’t remember you describing chlorophyll a in the methods….how did you measure this?!

line 250: Also, by including SOC with the nutrients, aren’t you also kind of answering objective 2 here? Like, is objective 2 still necessary after this?

line 252: Algae are often a major contributor of organic matter to the system, and likely related to C acquisition….could be just as important, or more important, than SOC!

Figure 3, panel d: This is up to the authors and editors, but to me it would be more clear to plot microbial C limitation and microbial N/P limitation separately rather than on the same figure…..right now it takes a while to understand the figure, and would probably be more clear in two

line 347: This would also be a reasonable place to report the average number of reads per sample, etc, although Im not seeing any such information here

Figure 5: Maybe the figure will look different once published, but for me, it is almost impossible to read the text in Figure 5 (other figures are also a bit difficult). Furthermore, in figures 4 and 5, colors are an important aspect of the figures, but it is not clear what they indicate until halfway through the caption. Perhaps at least for the bar graphs, there could be a legend that shows the meaning of the colors in the actual figure?

line 423: I would actually be curious how MML relates to chla, since it could also represent a carbon source, but also a highly labile one….did you look into that?

line 445: I mean, it is likely that N is limiting time to time in all of these ecosystems, but I think these days the opposite is actually more likely to be true…..In general, streams for example are thought to be more saturated with N than limited by them…. 

line 445: it doesn’t really inhibit AP activity….they more just don’t produce as much AP, no?

line 471: I agree that response mechanisms cannot be easily extrapolated from one study to the next. Even in this study, I think it is hard to squarely place the mechanism on salinity, which would be necessary to extrapolate it outward to other sites and systems…..

line 472: pH can also have quite strong effects on the sediments capacity to bind phosphorus, I think, which can alter its availability in the sediments/watercolumn

line 517: although the hypothesis was just that it would be associated, not really the direction of association....

line 529: maybe there could be other C sources than SOC as well?

line 563: but are these salinities not indigenous to these lakes you studied?

Reviewer 2 Report

Comments and Suggestions for Authors

This manuscript is well written.

Specific comments

Line 162: Please provide the country for Illumina Inc.

Reviewer 3 Report

Comments and Suggestions for Authors

The manuscript presents valuable information about the microbial communities in the inland lakes in the massive salinity gradient. The manuscript is well written, especially the introduction and discussion section. The results are well-presented, but I can’t fully evaluate them because of the lack of review supplementary materials. Thus, I would like to get those materials for further evaluation. Despite this, the presented results supported the well-written discussion and conclusion. I have a few comments, mainly on the method section and the statistical approach, which the authors will find in the attached revised PDF version of the manuscript.

Reviewer 4 Report

Comments and Suggestions for Authors

Thank you for giving me the opportunity to review the manuscript entitled “Microbial metabolic limitations and their relationships with sediment organic carbon across a lake salinity gradient in Tibetan Plateau“.

1.      The study investigates microbial metabolic limitations (MMLs) and their relationship with sediment organic carbon (SOC) across a salinity gradient in Tibetan Plateau lakes. The objectives are clearly stated, but the novelty is moderate.

2.      The introduction is well-structured; however, some sections (e.g., lines 36–66) could be condensed to improve readability.

3.      The experimental design is appropriate, with a comprehensive survey of 25 lakes. However, the following concerns should be addressed:

  • The enzyme assays should specify whether substrate concentrations were standardized across all samples.
  • The statistical analyses, particularly variance partitioning and path modeling, should clarify whether multicollinearity was checked among predictors.
  • The reproducibility of field sampling methods could be improved by describing specific precautions taken to minimize contamination beyond sterilization (line 111).
  • Include details on the exact calibration and controls for enzyme activity measurements.
  • Provide justification for using specific environmental parameters over others in statistical models.
  • Clarify how sediment pH and water salinity were measured over time to ensure consistency.

4.      The results are well-organized; however, some figures (e.g., Figure 3) contain excessive detail that could be simplified. Also, the results text should align more closely with tables and figures to avoid redundancy.

5.      The discussion appropriately interprets results in the context of existing literature. However:

  • It lacks a mechanistic explanation of why microbial carbon limitation increases under higher salinity beyond enzyme stoichiometry (lines 461–471).
  • The impact of microbial diversity on SOC should be better contextualized with additional references.
  • The discussion on environmental stressors (lines 563–574) could better distinguish between direct and indirect effects.

6.      The conclusions are supported by the results and align with the main research question. However:

  • The authors should clearly state the limitations of their study (e.g., temporal variability in salinity, potential unmeasured confounding factors).
  • The implications for ecosystem modeling could be expanded.

7.      The manuscript is generally well-written, but several minor grammatical issues should be corrected:

  • Line 36: change "Due to intensifying climate warming" to "Due to increasing climate warming."
  • Line 109: Modify "The equipment, tools, and containers used..." to "All equipment, tools, and containers used..."
  • Line 472: change "This speculation is plausible because of..." to "This hypothesis is plausible because of..."
  • Consistency in terminology (e.g., "microbial metabolic limitations" vs. "microbial enzyme limitations").

Round 2

Reviewer 4 Report

Comments and Suggestions for Authors

No more comments

Author Response

Thank you for your thoughtful review and positive feedback. We are glad to hear that the introduction provides sufficient background and includes all relevant references. We also appreciate your recognition that the methods were adequately described and the results clearly presented. 

Thank you again for taking the time to review this manuscript, which have made this manuscript more rigorous and complete.